# The PedAL/EuPAL Project: A Global Initiative to Address the Unmet Medical Needs of Pediatric Patients with Relapsed or Refractory Acute Myeloid Leukemia

**DOI:** 10.3390/cancers16010078

**Published:** 2023-12-22

**Authors:** Valeria Ceolin, Sae Ishimaru, Seth E. Karol, Francisco Bautista, Bianca Frederika Goemans, Gwenaëlle Gueguen, Marieke Willemse, Laura Di Laurenzio, Jennifer Lukin, Harm van Tinteren, Franco Locatelli, Arnaud Petit, Daisuke Tomizawa, Alice Norton, Gertjan Kaspers, Dirk Reinhardt, Sarah K. Tasian, Gwen Nichols, Edward Anders Kolb, Christian Michel Zwaan, Todd Michael Cooper

**Affiliations:** 1Princess Máxima Center for Pediatric Oncology, 3584 Utrecht, The Netherlands; v.c.ceolin@prinsesmaximacentrum.nl (V.C.); s.i.ishimaru@prinsesmaximacentrum.nl (S.I.); f.j.bautistasirvent@prinsesmaximacentrum.nl (F.B.); b.f.goemans@prinsesmaximacentrum.nl (B.F.G.); m.e.willemse@prinsesmaximacentrum.nl (M.W.); h.vantinteren@prinsesmaximacentrum.nl (H.v.T.); g.j.l.kaspers@prinsesmaximacentrum.nl (G.K.); 2Department of Pediatric Hematology/Oncology, Regina Margherita Children’s Hospital, University of Turin, 10126 Turin, Italy; 3Department of Oncology, St. Jude Children’s Research Hospital, Memphis, TN 38105, USA; seth.karol@stjude.org; 4The Innovative Therapies for Children with Cancer (ITCC) Consortium, 94805 Paris, France; 5Center of Clinical Investigations, INSERM CIC 1426, Robert Debre Hospital, University of Paris, 75019 Paris, France; gwenaelle.gueguen@aphp.fr; 6Leukemia & Lymphoma Society, Rye Brook, NY 10573, USA; laura.dilaurenzio@lls.org (L.D.L.); jennifer.lukin@lls.org (J.L.); gwen.nichols@lls.org (G.N.);; 7Department of Hematology/Oncology and Cell and Gene Therapy, IRCCS Bambino Gesù Children’s Hospital, Catholic University of the Sacred Heart, 00163 Rome, Italy; franco.locatelli@opbg.net; 8Department of Pediatric Hematology and Oncology, Hôpital Armand Trousseau, Assistance Publique Hôpitaux de Paris, APHP Sorbonne Université, 75012 Paris, France; arnaud.petit@aphp.fr; 9Division of Leukemia and Lymphoma, Children’s Cancer Center, National Center for Child Health and Development, Tokyo 104-0045, Japan; tomizawa-d@ncchd.go.jp; 10Birmingham Children’s Hospital, Birmingham B4 6NH, UK; alice.norton1@nhs.net; 11Emma Children’s Hospital, Amsterdam UMC, Vrije Universiteit, 1105 Amsterdam, The Netherlands; 12Department of Pediatric Hematology/Oncology, Pediatrics III, University Hospital of Essen, 45147 Essen, Germany; dirk.reinhardt@uk-essen.de; 13Division of Oncology, Children’s Hospital of Philadelphia, Department of Pediatrics, School of Medicine, Center for Childhood Cancer Research, University of Pennsylvania, Philadelphia, PA 19104, USA; tasians@chop.edu; 14Department of Pediatric Hematology/Oncology, Erasmus University MC-Sophia Children’s Hospital, 3015 Rotterdam, The Netherlands; 15Division of Hematology/Oncology, Seattle Children’s Hospital, University of Washington, Seattle, DC 98105, USA; todd.cooper@seattlechildrens.org

**Keywords:** acute myeloid leukemia, clinical trials, pediatric, refractory/relapsed, PedAL, EuPAL

## Abstract

**Simple Summary:**

The prognosis of children with relapsed/refractory (R/R) acute myeloid leukemia (AML) remains poor, and innovative treatments are needed. Most drugs approved for adults with AML over the last decade are not available or not licensed for use in children. Clinical trials in pediatric AML to assess the safety and/or efficacy of new drugs may take a long time to recruit given smaller patient numbers. The overarching aim of the Pediatric Acute Leukemia (PedAL) program, supported by the Leukemia and Lymphoma Society as part of its Dare to Dream Project, is to establish new standards of care for children with R/R AML via international collaboration in Europe, North America, and Asia-Pacific to accelerate clinical trial conduction, increase access to promising new therapies, and create a registry of uniformly-collected data. These efforts will facilitate biomarker-driven approaches of targeted therapies, of which an overview is provided in this manuscript, with the intent to obtain regulatory approvals.

**Abstract:**

The prognosis of children with acute myeloid leukemia (AML) has improved incrementally over the last few decades. However, at relapse, overall survival (OS) is approximately 40–50% and is even lower for patients with chemo-refractory disease. Effective and less toxic therapies are urgently needed for these children. The Pediatric Acute Leukemia (PedAL) program is a strategic global initiative that aims to overcome the obstacles in treating children with relapsed/refractory acute leukemia and is supported by the Leukemia and Lymphoma Society in collaboration with the Children’s Oncology Group, the Innovative Therapies for Children with Cancer consortium, and the European Pediatric Acute Leukemia (EuPAL) foundation, amongst others. In Europe, the study is set up as a complex clinical trial with a stratification approach to allocate patients to sub-trials of targeted inhibitors at relapse and employing harmonized response and safety definitions across sub-trials. The PedAL/EuPAL international collaboration aims to determine new standards of care for AML in a first and second relapse, using biology-based selection markers for treatment stratification, and deliver essential data to move drugs to front-line pediatric AML studies. An overview of potential treatment targets in pediatric AML, focused on drugs that are planned to be included in the PedAL/EuPAL project, is provided in this manuscript.

## 1. Introduction and Background

In part I, we introduce a summary on pediatric acute myeloid leukemia (AML) and the challenges related to its medical need. In part II, we present the PedAL/EuPAL collaboration, part of the Leukemia and Lymphoma Society’s “Dare to Dream” project. We further explain its aim and content and discuss the advantages of this large international collaboration which has the aim to facilitate the access of children with acute leukemia or myeloid malignancies to new drugs. In part III, we provide an overview of potential treatment targets in pediatric AML, focusing on drugs that are planned to be included in the PedAL/EuPAL project.

Acute myeloid leukemia accounts for approximately 20% of cases of acute leukemia in children and adolescents younger than 18 years of age. Overall survival (OS) rates for children with AML treated in contemporary clinical trials are now in the 70–80% range [1,2,3]. 

Improvements in survival have been achieved largely through the intensification of ‘conventional’ cytotoxic chemotherapy (cytarabine and anthracyclines), effective use of allogeneic hematopoietic stem cell transplantation (HSCT), improved risk stratification based upon AML-associated genetics and early response assessments, and improvements in supportive care [1,4]. However, relapses occur in 25–40% of patients [5]. 

In Western Europe, approximately 500 children with newly diagnosed AML are included in trials of the major study groups each year, and in the United States of America (USA) around 700 patients < 22 years old with newly diagnosed AML are enrolled annually in phase III clinical trials (Table 1). 

With a 25–40% relapse rate and accounting for mortality during active therapy, approximately 400 children included in the cooperative group clinical trial protocols will subsequently relapse. The survival for pediatric patients after a first relapse is at best 50% [6,7]. The median remission re-induction rate in a first relapse is close to 60% with a range of 35% to 81% (Table 2). 

There is a clear distinction in outcome between an early first relapse (within a year from initial AML diagnosis) and late relapse (more than a year from initial diagnosis) with the latter achieving higher survival rates [8]. Time to relapse is also linked to genetic risk groups, as standard/favorable-risk genetic patients tend to relapse later than patients with poor-risk genetics, who typically relapse within the first year. 

Of interest, it appears that the survival of patients in a first relapse has increased to almost 50% in the most recent era (2013–2017) [6,7]. Moreover, the number of patients transplanted as consolidation after a first relapse is increasing: in the AML 2001/01 trial, 69% of the 189 patients included in the first time period (from April 2001 to March 2005) were able to proceed to HSCT, compared to 82% of the 76 patients in the most recent August 2013–December 2017 period [6], demonstrating an improved ability to achieve remission and avoid severe complications during retrieval therapy, success in locating a suitable HLA-compatible donor, and the capacity to perform transplantation across the HLA barrier. This may at least in part explain why more patients are surviving, together with improved supportive care options.

A second relapse occurs in approximately 5–8% of all patients treated for newly diagnosed AML, and in 20–45% of patients with a first relapse of AML [6,7,14,22]. The five-year OS for patients in a second relapse is appreciably worse at 15–20% [14]. Early second relapse (defined as a second relapse within one year after a first relapse) remains a predictor of particularly dismal outcomes with a five-year OS of only 2% [22].

AML refractory to initial induction therapy (primary chemorefractory) occurs in approximately 3–7% of patients [12,22]. However, the rate of poor reinduction chemotherapy response is higher at relapse. In the Children’s Oncology Group (COG) report, which included data from children relapsing after treatment on COG AAML0531 and COG AAML1031, 29% of patients had greater than 5% blasts at the end of one cycle of re-induction therapy, meeting the protocol definition of refractory disease. Their 5-year OS was only 24% [6,7]. In the Berlin–Frankfurt–Münster Study Group (BFM SG) first-relapse cohort, including patients treated after closure of the last published AML-BFM 2001/01 trial [12], 21% of patients receiving treatment with two re-induction cycles (fludarabine/cytarabine versus fludarabine/cytarabine with liposomal daunomycin) did not respond. These patients had a dismal survival with a 5-year OS of only 28%. In the Nordic Society of Pediatric Hematology and Oncology–Dutch Belgian (NOPHO-DB) retrospective cohort of patients collected by White et al. [14], the one-year OS was 25% for patients with a refractory first relapse (n = 98 out of 277 of the patients who experienced a first relapse, representing 35% of all first relapses). 

### The Need to Establish a Standard of Care for Relapsed Patients

Contrary to newly diagnosed patients, fewer than 20–25% of patients at relapse are treated in a study due to a lack of access or lack of clinical trials themselves [5,12,23]. Since the AML BFM 2001/01 protocol [12], there has been no international standard of care (SOC) establishing trials in children with AML. Allogenic HSCT in second remission is the therapeutic standard for all relapsed patients [24] and is usually applied after one to two blocks of intensive reinduction therapy. Primary refractory and relapsed AML are usually treated as one single clinical entity [25]. 

Though anthracycline-containing regimens generally induce better response rates, especially in patients with core binding factor fusion AML, not all patients can tolerate additional anthracyclines at relapse due to a high prior cumulative exposure or evidence of cardiac dysfunction during the first-line therapy [12]. Given the concern for long-term cardiac toxicity with high cumulative anthracycline dosing, most collaborative groups have tried to reduce anthracycline exposure and/or use cardioprotective agents such as dexrazoxane or liposomal anthracycline formulations at relapse. Liposomal anthracyclines have shown evidence of reduced cardiotoxicity [26,27]. In Europe, liposomal daunorubicin has been used off-label as the preferred anthracycline at relapse [1]. While SOC for first-relapse patients in Europe (liposomal daunorubicin-FLA) was established in the AML BFM 2001/01 protocol by Kaspers et al. [12], liposomal daunorubicin is no longer commercially available. In the USA, patients have historically been treated with high-dose cytarabine or FLA with or without anthracyclines, such as daunorubicin or mitoxantrone. More recently, a liposomal formulation of cytarabine and daunomycin agent CPX-351 was tested in a phase I/II study of children with first-relapse AML and reported by Cooper et al. [16]. This trial led to a label extension of CPX-351 in North America to include children aged 1 year and older with therapy-related AML (t-AML) or AML with myelodysplasia-related changes (AML-MRC), but it is not licensed in Europe.

Patients with a second relapse or who are refractory (R/R) to the second line of treatment have thus far been considered candidates for experimental therapy approaches in early-phase clinical trials. Moreover, some groups consider patients with an early first relapse or specific high-risk genetic alterations also eligible for phase I/II studies given their poor outcome. Taken together, despite successes, there clearly remains a significant unmet medical need to develop more effective and less toxic treatments, especially for R/R patients. Furthermore, as outcome results in first-relapse AML have slightly improved over time, there may be room to define a SOC approach for second relapse, especially for patients who still have options for HSCT (or second HSCT).

Innovative treatments with new mechanisms of action to eradicate resistant disease in all stages are required. Recent large-scale discovery efforts, such as next-generation sequencing methods, have identified recurrent mutational, signaling pathway, and cell surface antigen targets for new therapies. Innovative oncology drugs with new mechanisms of action are available today for adults with AML (as summarized in Table 3). However, access to innovative therapies remains insufficient and slow for children and adolescents [23]. Given the clinical and biological differences between adult and pediatric AML, there is also a need to test molecules developed specifically for pediatric AML biology.

## 2. The PedAL/EuPAL Project

Precision medicine for adult patients with AML is the goal of the Beat AML precision Master Trial [28], a collaborative clinical trial launched by the Leukemia and Lymphoma Society (LLS) in the USA in 2016, which uses advanced genomic technology to identify each adult AML patient’s cancer-driving genetic mutations, and then allocates patients to the most promising targeted treatment. It has effectively convened researchers and physicians, regulatory agencies, pharmaceutical and biotech partners, and patient groups [28]. Although there are many novel medicinal products being evaluated in adults with AML, there are factors that make the clinical development of adult AML drugs in children difficult. For example, targeted agents for adults are often not applicable in children because of different genetic abnormalities, different tolerability to new drugs between children and adults, or the rarity of AML in children that makes the enrolment a challenge. The LLS Dare to Dream Project (Dare to Dream Project: Treating Kids with Blood Cancer|LLS) is working to translate the successes of Beat AML into a novel international clinical trial program with the aim to accelerate the development of new treatments and break down barriers to care for children with acute leukemia. 

The current regulatory incentives, such as those included in the Pediatric Regulation in Europe or the Research to Accelerate Cures and Equity (RACE) for Children Act in the USA, have facilitated the development of agents in the pediatric population but are clearly still insufficient, and new incentives are needed [29]. However, the current landscape is scattered with multiple, competing, relatively small trials conducted in silo by different sponsors and targeting the same rare patient populations. In addition, there is often a lack of harmonization of diagnostic tests, inclusion and exclusion criteria, and outcome parameters across studies, so it is difficult to compare these trials to each other. 

As the number of children with R/R AML available for studies is limited and pediatric AML is genetically and immunophenotypically heterogeneous, prioritization needs to take place in the selection of agents for clinical trial investigation. The evaluation of these treatment strategies will require collaboration between cooperative groups to ensure that adequate numbers of patients to which each therapy is directed can be studied. To help achieve such goals, the ACCELERATE Pediatric Strategy Forum (PSF) was created to evaluate science, facilitate dialogue, and provide an opportunity for constructive interactions amongst relevant stakeholders, including patient advocates, pediatric oncologists, biotechnology/pharmaceutical companies, and government regulators [30]. In April 2019, the fourth PSF was organized by ACCELERATE in collaboration with the European Medicines Agency (EMA) with the participation of the Food and Drug Administration (FDA) with a goal of facilitating prioritization of and access to innovative medicines for the treatment of children and adolescents with AML [23]. The main conclusions of this meeting were that efforts must be coordinated to complete the enrollment of patients in studies of *FLT3* inhibitors and to prioritize the pediatric development of CD123-targeting drugs. For AML with rare mutations that are more frequent in adolescents than in younger children, adult trials should also enroll adolescents and, when scientifically justified, efficacy data could be extrapolated. It was considered important to achieve the international standardization of methodologies and definitions of measurable residual disease (MRD). 

In this context, the LLS ‘Dare to Dream Pediatric Acute Leukemia (PedAL)’ project is a strategic initiative that aims to overcome the obstacles in treating children with acute leukemia and myeloid malignancies via a transatlantic/Asia-Pacific global collaboration, including countries from Europe, Israel, North America, and Asia-Pacific (e.g., Japan, Australia, and New Zealand). Organizations such as COG, the Innovative Therapies for Children with Cancer consortium (ITCC), and the European Pediatric Acute Leukemia (EuPAL, including the United Kingdom) Foundation, set up to coordinate the European efforts for the PedAL initiative, are involved in this project. The aim is to run a program of early- and late-phase clinical trials evaluating the safety and the efficacy of new agents in pediatric leukemia and myeloid malignancies, eventually setting a new standard of treatment for first- and second-relapsed AML. PedAL/EuPAL have also already created contractual and infrastructure elements that will facilitate future international clinical trials.

### 2.1. Screening Trial and Registry

In North America, Australia, and New Zealand, patients can enroll in the APAL2020SC PedAL screening protocol (NCT04726241), which utilizes commercial laboratories (Hematologics, Inc., Seattle, WA; Foundation Medicine, Inc, Cambridge, Massachusetts) for standardized genetic and immunophenotyping diagnostics. In Europe, the project is implemented differently, and the current project consists of a registry protocol (the EuPAL2021 Registry) collecting data from the national reference laboratories. There is also a specific project dedicated to the harmonization of flow cytometric and genetic/molecular diagnostics, including the country/collaborative group-specific centralized flow cytometry/genetic laboratories in Europe and both Hematologics and Foundation Medicine in North America. 

Any patient with known or suspected R/R AML who is less than 22 years of age is eligible for inclusion in the registry in Europe or in APAL2020SC in North America, Australia, and New Zealand to provide confirmation of relapse and target expression using standardized methodologies. After confirmation of relapse, patients may be enrolled, at their or the physician’ s discretion, on a PedAL sub-trial, another (unrelated) clinical trial, or be treated with standard-of-care chemotherapy approaches. 

In Europe, the project consists of a master protocol with sub-trials, the ITCC-101 PedAL/EuPAL Master protocol, implementing a stratification approach to allocate patients to the planned sub-trials. The assignment of patients to sub-trials depends on existing clinical data in other populations, biomarker prevalence (such as *KMT2A* rearrangements, high MRD prior to HSCT), disease stage (first relapse, second/greater relapse, refractory), eligibility criteria, and slot availability. This interventional platform trial allows flexibility in inclusion of different sub-trials and addition or exclusion of investigational medicinal products (IMPs) or patient populations enabling an efficient transition of IMPs to a confirmatory trial. Each sub-trial will effectively run as one study performed on a global scale, and data for each sub-trial will be entered in one study-specific database. However, some studies may only be implemented in one continent as they are small studies or implemented outside the ITCC-101 PedAL/EuPAL Master protocol for a variety of reasons. 

### 2.2. The First Sub-Trial Opened: APAL2020D

ITCC-101/APAL2020D is an open-label randomized phase III trial of fludarabine, high-dose cytarabine (FLA) + gemtuzumab ozogamicin (GO), with or without venetoclax, in children in a second relapse of their AML or in a first relapse in patients who can no longer tolerate anthracyclines (NCT05183035, EudraCT number 2021-003212-11) [31]. The APAL2020D study enrolls children, adolescents, and young adults with AML without *FLT3*-*ITD* mutation who are sufficiently fit to undergo intensive chemotherapy. There are two cycles of treatment: Cycle 1 consists of FLA + GO and Cycle 2 consists of FLA only (Figure 1). Patients are randomized to receive this chemotherapy with or without venetoclax for both cycles. Responding patients after the first two cycles can undergo allogeneic HSCT. Patients who have responded to protocol therapy but who cannot tolerate HSCT may receive maintenance therapy consisting of azacitidine or the combination azacitidine-venetoclax as per the randomized arm. 

The primary objective is to compare the OS of venetoclax in combination with FLA + GO compared with FLA + GO alone. The study hypothesizes that the addition of venetoclax will improve 1-year survival by approximately 20%. Ninety-eight randomized patients are required (80% power, one-sided alpha of 2.5%) to increase survival from 47% to 68%. Secondary objectives include the assessment of response, safety, pharmacokinetics, and rate of transition to HSCT. The morphology and MRD by flow cytometry assessments will be centralized.

The LLS PedAL Initiative, Limited Liability Company (LLC), is the sponsor in the USA, Canada, Australia, and New Zealand. The Princess Máxima Center is the sponsor in Europe, the United Kingdom, Israel, and Switzerland. The study opened in June 2022 with recruitment ongoing.

## 3. Potential Treatment Targets in Pediatric Acute Myeloid Leukemia

In this section, we will show the potential treatment targets in pediatric AML, with a focus on those for which there are active and developing clinical trials under the ITCC-101 PedAL/EuPAL Master protocol. An overview of potential treatment targets in pediatric AML is given in Figure 2. 

Cell therapy, recently summarized in other papers [32,33], is not discussed in this manuscript as, at the moment, there are no sub-trials planned in the PedAL/EuPAL Project. 

**Figure 2 cancers-16-00078-f002:**
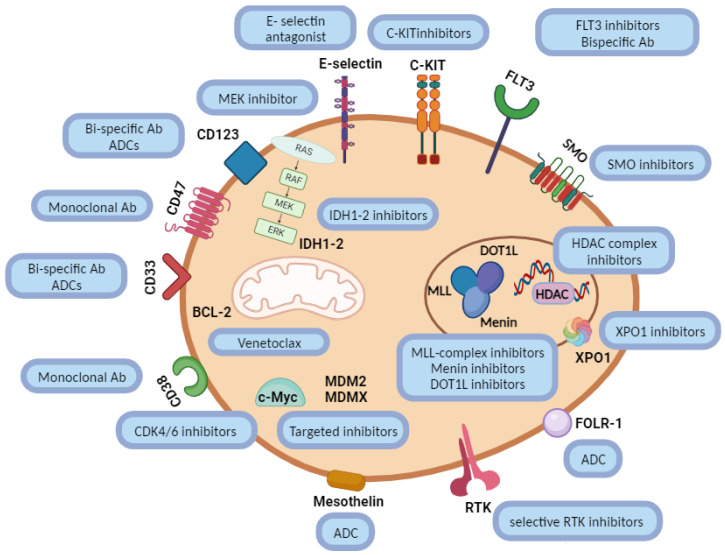
Overview of current treatment targets in acute myeloid leukemia. Ab = antibody; ADC = antibody–drug conjugate; RTK = receptor tyrosine kinase. Note that cellular therapy initiatives are not included. Adapted from Brivio et al. [34] and Cooper et al. [33].

### 3.1. Novel Drugs in Pediatric AML

#### 3.1.1. Gemtuzumab Ozogamicin

Gemtuzumab ozogamicin (GO), a CD33-targeting antibody–drug conjugate (ADC), has improved outcomes in a major subgroups of patients with AML. In the COG phase III pediatric randomized controlled front-line trial (COG AAML0531), GO was added to induction and intensification. The incorporation of GO into therapy was associated with a better EFS but with no difference in OS between arms [35]. GO is incorporated into the ongoing COG phase III front-line trial (COG AAML1831; NCT04293562) as part of backbone therapy. It is also being evaluated in newly diagnosed patients in the United Kingdom, Europe, Australia, and New Zealand in the MyeChild 01 phase II/III trial (NCT02724163). This study compares two different schedules of administration of GO. As explained above, this ADC is also part of the PedAL/EuPAL sub-trial on venetoclax (NCT05183035).

The USA and Europe have approved GO for different pediatric indications: for treatment of patients 2 years and older with R/R CD33-positive AML in the USA and for patients aged 15 years and above with newly diagnosed AML in Europe. This difference in regulatory status poses challenges to the conduct of transatlantic clinical trials.

#### 3.1.2. *FLT3* Inhibitors

*FLT3* tyrosine kinase domain (TKD) mutations or the mutant *FLT3*-internal tandem duplications (*FLT3*-ITD) are, with approximately 15–20% of all cases, frequent cooperating lesions in pediatric AML, although the kinase domain mutations are probably not prognostically significant [36,37,38]. *FLT3*-ITD is associated with a poor outcome, especially in case of a high allelic ratio [36,37,38]. 

In adults, three *FLT3* inhibitors have been authorized for use by the FDA, including midostaurin in newly diagnosed patients in combination with chemotherapy; gilteritinib as a monotherapy for relapsed AML; and quizartinib, recently approved, with standard cytarabine and anthracycline induction and cytarabine consolidation, and as a maintenance monotherapy following consolidation chemotherapy, for the treatment of adult patients with newly diagnosed *FLT3-ITD*-positive AML [39,40,41]. 

Studies with various *FLT3* inhibitors are planned or ongoing in children and have made their way into front-line therapy:-Midostaurin in combination with chemotherapy is being studied in newly diagnosed children in an ongoing company-sponsored study (NCT03591510);-Gilteritinib is currently investigated in children with both newly diagnosed and R/R *FLT3-ITD* and/or *TKD* mutated AML (NCT04293562; NCT04240002);-Quizartinib for newly diagnosed *FLT3-ITD/NPM1*-wild type AML will soon be open to enrolment (EudraCT: 2022-002886-14), based on data from the ongoing relapse study (NCT03793478).

The recently completed COG AAML1031 trial studied the feasibility and efficacy of adding sorafenib to standard chemotherapy and as a single-agent continuation therapy. Data demonstrated that sorafenib dosing of 200 mg/m^2^/day was tolerable in conjunction with conventional chemotherapy, significantly improved EFS, relapse risk, and disease-free survival. The risk of relapse was approximately two-fold higher in high-allelic-ratio *FLT3*/ITD+ patients (n = 76) who did not receive sorafenib [42].

Recent studies have also demonstrated the benefit of post-HSCT maintenance therapy with sorafenib and midostaurin [42,43]. However, Mark Levis et al. [44] showed that there is no statistically significant improvement of relapse-free survival (RFS) in patients with *FLT3-ITD* AML treated with gilteritinib as a maintenance therapy following HSCT. However, there was a clinical improvement of RFS in a subgroup of patients with detectable MRD compared to patients without detectable MRD. 

### 3.2. Current High-Priority Pediatric AML Targets

#### 3.2.1. BCL-2 Inhibitors

Venetoclax is a BCL-2 inhibitor approved by the FDA and EMA in combination with azacitidine, decitabine, or low-dose cytarabine for newly-diagnosed AML in adults 75 years or older or who have comorbidities precluding intensive induction chemotherapy [45]. 

In the pediatric setting, venetoclax was tested in an industry-supported phase I study, the results of which have not yet been reported (NCT03236857) [46]. This trial aimed to evaluate the safety and pharmacokinetics of venetoclax monotherapy and/or in combination with chemotherapy, to determine the recommended phase II dose (RP2D) of venetoclax, and to assess the preliminary efficacy of the drug in pediatric and young adult participants with R/R malignancies [47]. The multicenter VENAML trial (NCT03194932) subsequently assessed the safety and activity of venetoclax in combination with high-dose cytarabine and idarubicin in pediatric patients with R/R AML [21]. This study also identified the RP2D of venetoclax in combination with high-dose chemotherapy. Venetoclax is now being studied in a randomized phase III trial with FLA/GO as a backbone in pediatric patients with AML in second relapse or in first-relapse patients that cannot tolerate anthracyclines (NCT05183035). It is the first sub-trial open under the ITCC-101 PedAL/EuPAL Master protocol (see the specific section above), and part of a regulatory commitment by the market authorization holder. 

#### 3.2.2. Menin Inhibitors

*KMT2A* rearrangements are one of the most common molecular subtypes of childhood AML, present in approximately 25% of patients with newly-diagnosed disease [7]. The outcomes of children with *KMT2A*-rearrangement vary depending upon the specific 5′ fusion partner with particularly poor outcomes in those with KMT2A::AFF1 resulting from the t(4;11)(q21;q23) translocation, KMT2A::MLLT4 from t(6;11)(q27;q23), KMT2A::ABI1 from t(10;11)(p11.2;q23), KMT2A::MLLT1 from t(11;19)(q23;p13.3), and KMT2A::MLLT10 from t(10;11)(p12;q23), with OS rates below 50% [48,49,50]. A major mechanism by which *KMT2A* fusion proteins maintain leukemogenesis is through interaction with chromatin-associated protein complexes. One of the critical proteins found in these complexes is menin (MEN1). The inhibition of the MEN1-KMT2A interaction is a potential therapeutic strategy for *KMT2A*-rearranged leukemia that may help to overcome the chemoresistance and poor clinical outcomes experienced by patients. Preclinical and clinical evidence shows that *NUP98*-rearranged, *NPM1*-mutated, and other high *MEIS1*-expressing AML subtypes may also be targetable with menin inhibition [51,52].

Ziftomenib (KO-539) is under evaluation in the KOMET-001 phase I/IIa clinical trial (NCT04067336) in adult patients with R/R AML [53]. The phase Ia dose-escalation study enrolled 30 adult patients with R/R AML regardless of genotype. Two dose-limiting toxicities occurred: pneumonitis and differentiation syndrome. The phase Ib clinical efficacy was dose-dependent. Composite CR was 33% with 75% MRD negativity [54]. Other molecules targeting the KMT2A–menin interaction are currently beingtested in adult patients, but no preliminary results have been reported to date (Daiichi Sankyo, NCT04752163; Janssen, NCT04811560; Biomea Fusion, NCT05153330; Sumitomo Dainippon Pharma, NCT04988555).

Several studies are planned or ongoing in children:-Revumenib (SNDX-5613) is being studied as a monotherapy in the company-sponsored AUGMENT-101 trial, a phase I/II trial that includes both adults and children with R/R *KMT2A*-rerarranged or *NPM1*-mutant acute leukemias (NCT04065399). Therapy with revumenib has been associated with a low frequency of Grade 3 or higher treatment-related adverse events and a 30% (18 of 60 evaluable patients) rate of complete remission (CR) or complete remission with incomplete hematologic recovery (CRi) [49]. Specific safety issues consist of QTc prolongation and an interaction with the cytochrome P450 3A4 inhibitor (CYP3A4), resulting in an RP2D with (Arm B) and without (Arm A) a strong CYP3A4 of 226 mg q12h and 276 mg q12h in Arm A and 113 mg q12h and 163 mg q12h in Arm B;-The AUGMENT-102 is a phase I, dose-escalation study also designed to evaluate the safety, tolerability, and preliminary anti-leukemic activity of revumenib (SNDX-5613) in combination with chemotherapy in patients with R/R leukemias harboring alterations in *KMT2A/MLL*, *NPM1*, and *NUP98* genes (NCT05326516);-Revumenib (SNDX-5613) is also being investigated in a phase I/II trial with decitabine/cedazuridine (ASTX727) and venetoclax in R/R AML or mixed-phenotype acute leukemia (MPAL) with a myeloid phenotype;-Ziftomenib will be studied in the second sub-trial that will be soon opened under the ITCC-101 PedAL/EuPAL Master protocol (APAL2020K, EudraCT number: 2023-505262-28-00).

### 3.3. Other Targets

#### 3.3.1. FOLR1-Targeted Agents

FOLR1 is a uniquely overexpressed cell surface target protein *CBFA2T3::GLIS2* AML, and FOLR1-expressing AML is sensitive to the FOLR1-directed ADC STRO-002 [55]. Between August 2021 and July 2022, 16 pediatric patients with R/R *CBFA2T3::GLIS2* AML received STRO-002 on a compassionate use basis, 10 as monotherapy and 6 in combination with chemotherapy (fludarabine/cytarabine, decitabine, methotrexate, or dasatinib). All 16 patients were evaluable for response; the best observed responses included 7 CRs with 6 MRD-negative remissions. STRO-002 was well-tolerated as both a monotherapy agent and in combination with cytotoxic agents [56]. 

ELU001 is a new chemical entity described as a C’Dot drug conjugate (CDC), consisting of payloads (exatecans) and targeting moieties (folic acid analogs) covalently bound by linkers to the C’Dot particle carrier. A dose escalation study to evaluate the safety and tolerability of ELU001 in pediatric patients who have relapsed and/or refractory *CBFA2T3::GLIS2* positive AML is active, but not yet recruiting (NCT05622591), and will be performed in the TACL consortium (T2022-001). 

#### 3.3.2. E-Selectin Inhibitors

GMI-1271 (uproleselan) is a novel E-selectin antagonist that disrupts the activation of cell survival pathways in preclinical models of AML [57]. GMI-1271 in combination with myelosuppressive chemotherapy was safe and showed promising anti-leukemic activity in adults with untreated or R/R AML [58]. GMI-127 is currently undergoing evaluation in combination with low-intensity agents for newly diagnosed AML in adults unfit for intensive chemotherapy (NCT04964505). The COG PEPN2113 phase I study is evaluating the safety and toxicity of uproleselan in pediatric patients with second or greater relapse of E-selectin ligand AML, MDS, or mixed-phenotype acute leukemia (NCT05146739). 

#### 3.3.3. IDH Inhibitors

Ivosidenib, enasidenib, and olutasidenib are isocitrate dehydrogenase (IDH) inhibitors. In adult patients, IDH1 mutations are associated with inferior outcomes, while IDH2 mutations have a more favorable prognosis [59]. Due to the rarity of IDH mutations in childhood AML [60] and the relatively good prognosis of this genetic subtype of leukemia, the safety and clinical activity of these agents have been challenging to establish in children. A safety and efficacy signal of enasidenib monotherapy is currently evaluated in children with R/R IDH2-mutated AML by the COG (NCT04203316), and in children with R/R malignancies through the eSMART platform trial in Europe (EudraCT nr. 2016-000133-40).

#### 3.3.4. XPO1 Inhibitors

A phase I trial for children with R/R acute leukemia of selinexor, an exportin 1 (XPO1) inhibitor, with FLA chemotherapy has demonstrated a manageable toxicity profile, and 7/15 (47%) of patients obtained CR or CRi during therapy [61]. The tolerability and efficacy of selinexor and venetoclax with and without chemotherapy are currently being explored further in children with R/R acute leukemias via the multi-site SELCLAX phase I trial (NCT04898894).

#### 3.3.5. CD123 Targeting

CD123, the interleukin-3 (IL-3) receptor alpha-chain, is expressed on the surface of leukemic cells in >80% of adult and childhood AML, and importantly is expressed on leukemia-initiating cells (LICs) [62,63]. It is also widely expressed in B-ALL. The expression of CD123 by T-ALL blasts is lower, with the exception of the more immature early thymic precursor (ETP) subtype. LICs are thought to be a major driver in relapse because they are relatively quiescent and chemo-resistant [62]. Testing of diagnostic and relapse samples show that CD123 expression is fairly stable in leukemic relapse samples including the LIC compartment [63]. High CD123 expression in pediatric AML is associated with high-risk *KMT2A* rearrangements and *FLT3-ITD* mutations and is independently prognostic of inferior outcomes [10,64]. The COG PEPN1812 phase I trial (NCT04158739) tested the safety and tolerability of flotetuzumab in children and adolescents/young adults with second or greater R/R AML via a rolling six design. Sixteen subjects (ages: 3–19 years) with R/R AML were enrolled from January 2020 to May 2021 with fifteen subjects receiving at least one dose. Flotetuzumab was determined to be safe and tolerable at the RP2D of 500 ng/kg/day (Dose Level (DL) 1) with an ORR of 20% [64]. However, the development of flotetuzumab was discontinued.

An open-label, dose-escalation/expansion study of SAR443579, a novel CD123 targeted natural killer cell engager, administered as a single agent in 12-year-old and older patients with R/R AML, B-ALL, or high risk-myelodysplasia (HR-MDS) is ongoing (NCT05086315) [], and further pediatric-specific testing of SAR443579 is planned.

## 4. Conclusions and Future Perspectives

The PedAL/EuPAL initiative is a global project with the goal of establishing new standards of care for children with R/R AML to improve the outcome and increase the long-term survival of these patients. Using a platform design for clinical trials is a promising approach for development of innovative medicines for the treatment of children and adolescents with AML.

The continued evaluation of targeted strategies by the pediatric AML community through recent international collaborations that allow the conduction of global trials to better study the growing number of rare molecular subsets, must be implemented with the identification of new predictive biomarkers of treatment response versus failure to determine which drugs should be prioritized. Despite many existing challenges, new biologic understanding and access to new chemotherapies and targeted therapies will make it possible to cure more children with AML.

## Figures and Tables

**Figure 1 cancers-16-00078-f001:**
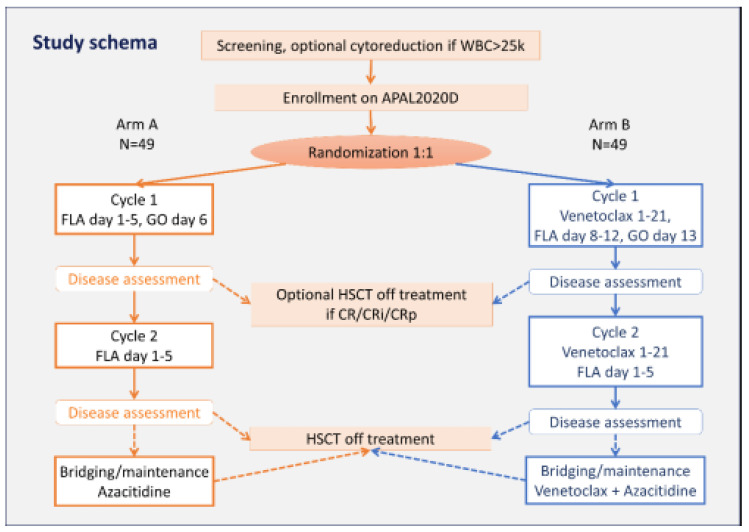
APAL2020D treatment schema.

**Table 1 cancers-16-00078-t001:** Annual enrollment of pediatric and adolescent/young adult patients with newly diagnosed AML enrolling in cooperative group clinical trials.

Study Group	Countries	New Patients per Year
NOPHO-DB-SHIP ^1^	Belgium, Denmark, Estonia, Finland, Hong-Kong, Israel, Iceland, Latvia, Lithuania, The Netherlands, Norway, Portugal, Spain, Sweden, Uruguay	~150
AIEOP-BFM ^2^	Austria, Czech Republic, Germany, Italy, Poland, Slovenia, Slovakia	~200
MyeChild	Australia, France, Ireland, Switzerland, United Kingdom	~180
JCCG ^3^	Japan	~100
COG ^4^	United States, Canada	~700
St. Jude	United States	~50
All groups		~1380

Data from ACCELERATE Platform (accelerate-platform.org, accessed on 2 November 2023). ^1^ NOPHO-DB-SHIP: Nordic Society of Pediatric Hematology and Oncology–Dutch Belgian–Spain–Hong-Kong–Israel–Portugal; ^2^ AIEOP-BFM: Italian Pediatric Hematology Oncology Association–Berlin–Frankfurt–Münster; ^3^ JCCG: Japan Children’s Cancer Group; ^4^ COG: Children’s Oncology Group.

**Table 2 cancers-16-00078-t002:** Results of recent studies for relapsed and refractory AML.

Study	Years of Enrollment	Study Question	Relapse Status	Nr. Pts	CR (%)	EFS (%) and (Median FU)	OS (%) and (Median FU)	Ref.
LAME89/91	1988–1998	Post-relapse survival	1st relapse	106	71	DFS 45 (5-y)	33 (5-y)	[8]
NOPHO	1993–2012	Various regimens	1st relapse	208	70		39 (5-y)	[9]
Japanese AML Cooperative Study	2000–2005	Various regimens	1st relapse after AML99 protocol	71	50		37 (5-y)	[10]
AML BFM-SG compassionate use series	1995–2014	GO+ cytarabine	Relapsed/refractory (mostly 2nd)	76			18 (all pts) 27 (transplanted) (4-y)	[11]
I-BFM AML 2001/01	2001–2009	FLAG vs. FLAG + DNX	1st relapse	394	59 vs. 69		34 (5-y)	[12]
AML-BFM SG relapse registry	2009–2017	DNX+FLA(G) +/− FLA vs. other re-induction therapies	1st relapse	197	45 n = 156 DNX- FLA(G) +/− FLA	30 (5-y) n = 156 DNX-FLA(G) +/− FLA	42 (5-y)	[7]
I-BFM AML 2001/02 phase II	2001–2009	GO single agent	Relapsed/ refractory	30	ORR 37		27 (3-y, responders only)	[13]
NOPHO-DB SHIP	2004–2019	Retrospective cohort review	Refractory 1st relapse/ 2nd relapse	157			22 (1-y) 14 (5-y)	[14]
JPLSG	2006–2010	Retrospective cohort review	1st relapse	111	64 63 ECM 66 FLAG 36 others		36 (5-y)	[15]
COG AAML0523 phase I/II	2007–2012	Clofarabine/cytarabine	Relapsed/ refractory	51	ORR 48		46 (3-y, responders only)	[16]
COG AAML-07P1 phase II	2008–2011	Cytarabine/ida/bortezomib vs. Cytarabine/etoposide/bortezomib	Relapsed/ refractory tAML	37	57 vs. 48		39 (2-y)	[17,18]
ITCC 020 phase Ib	2010–2014	Clofarabine with FLA + DNX	Relapsed/ refractory	34	ORR 68	35 (1-y)	50 (1-y)	[19]
COG AAML1421 phase I/II	2016–2018	CPX351, followed by FLAG	Relapsed/ refractory	37	ORR 81			[20]
Venetoclax phase I	2017–2019	Venetoclax/cytarabine	Relapsed/ refractory	38	ORR 69			[21]

AML = acute myeloid leukemia; BFM SG = Berlin–Frankfurt–Münster Study Group; COG = Children’s Oncology Group; CR = complete remission; DFS = disease-free survival; DNX = liposomal daunorubicin; ECM = etoposide, cytarabine, and mitoxantrone; EFS = event-free survival; FLA(G) = fludarabine, cytarabine, with or without GCSF; FU = follow-up; GO = gemtuzumab ozogamicin; I-BFM = International BFM study group; ida = idarubicin; ITCC = Innovative Therapies for Children with Cancer consortium; JPLSG = Japanese Pediatric Leukemia/Lymphoma Study Group; n = number; NOPHO = Nordic Society of Pediatric Haematology and Oncology; ORR = overall response rate (usually comprising CR plus CRp, CRi); OS, overall survival; pts = patients; R/R = relapsed/refractory; Ref = reference; tAML = therapy-related AML; y = year.

**Table 3 cancers-16-00078-t003:** Drugs approved for adult AML and status for pediatric AML.

Drug	Mechanism of Action	Indication	Approval Status	Pediatric Studies	Pediatric Study Population
CPX-351	Liposomal formulation cytarabine/daunorubicin	Newly diagnosed t-AML or AML-MRC	USA: approved for adult and pediatric patients aged 1 and older EU: approved for adult patients	EudraCT 2020-000142-34 [recruiting] NCT04915612 [recruiting] NCT04293562 [recruiting]	early 1st R or ≥2nd R AML 1–21 y relapsed AML ≤21 y newly diagnosed AML <21 y
Enasidenib mesylate	IDH2 inhibitor	R/R AML with *IDH2* mutation	USA: approved for adult patients	NCT04203316 [recruiting] NCT02813135 [recruiting]	*IDH2* mutated ≥2nd R/R AML 2–18 y R/R hematologic or solid tumor <18 y
Ivosidenib Olutasidenib	IDH1 inhibitor	Newly diagnosed *IDH1* mutated patients with AML R/R AML with *IDH1* mutation R/R AML with *IDH1* mutation	USA: approved for adult patients EU: approved for adult patients	NCT03245424 [expanded access program]	R/R *IDH1* mutated AML ≥12 y
Gemtuzumab Ozogamicine	Conjugated antibody against CD33	Newly diagnosed CD33-positive AML in adults and R/R CD33-positive AML in adults and in pediatric patients 2 years and older	USA: approved for adult and pediatric patients aged 2 and older EU: approved for patients aged 15 years and above who have not tried other treatments.	NCT04293562 [recruiting] NCT02724163 [recruiting]	newly diagnosed AML ≤21 y AML/HR MDS /isolated MS <18 y
Gilteritinib	*FLT3* inhibitor	R/R AML with *FLT3* mutation as monotherapy	USA: approved for adult patients EU: approved for adult patients	NCT04293562 [recruiting] NCT04240002 [recruiting]	newly diagnosed AML ≤21 y *FLT3*-ITD R/R AML ≥6 m–21 y
Midostaurin	*FLT3* inhibitor	Newly diagnosed AML with *FLT3* mutation, combined with chemotherapy	USA: approved for adult patients EU: approved for adult patients	NCT03591510 [recruiting]	*FLT3*-mutant AML ≥3 m–17 y
Glasdegib	SMO inhibitor hedgehog pathway	Newly diagnosed AML in patients ≥ 75 years or ineligible for intensive chemotherapy	USA: approved for adult patients EU: approved for adult patients	Waiver for pediatric development	
Venetoclax	BCL2 inhibitor	Newly diagnosed AML in patients ≥ 75 years or ineligible for intensive chemotherapy, in combination with HMA/low dose Ara-c	USA: approved for adult patients EU: approved for adult patients	NCT05183035 [recruiting]	≥2nd R AML or 1st R AML intolerant to anthracycline ≥ 29 d−≤ 21 y
Azacitidine	Hypomethylating agent	Maintenance and adult AML and MDS	USA: approved for adult patients and for children with JMML EU: approved for adult patients	NCT03825367 [closed early due to lack of activity]	R/R AML 1–30 y

AML = acute myeloid leukemia; AML-MRC = AML with myelodysplasia-related changes; d = day; EU = Europe; HMA = hypomethylating agent; HR = high risk; JMML = juvenile myelomonocytic leukemia; m = months; MDS = myelodysplasia; MS = myeloid sarcoma; R/R = relapsed/refractory; t-AML = therapy-related acute myeloid leukemia; USA = United States of America; y = years.

## Data Availability

No new data were created or analyzed in this study. Data sharing is not applicable to this article.

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
