# Peer review of "The PedAL/EuPAL Project: A Global Initiative to Address the Unmet Medical Needs of Pediatric Patients with Relapsed or Refractory Acute Myeloid Leukemia"

_cancers, 2023, doi:10.3390/cancers16010078_

Round 1

Reviewer 1 Report

Comments and Suggestions for Authors

In their manuscript “The PedAL/EuPAL project: a global initiative to address the unmet medical needs of pediatric patients with relapsed or refractory acute myeloid leukemia”, the authors describe a global-scale project aiming to establish sufficient cohorts of pediatric patients with relapsed or refractory acute myeloid leukemia (AML). The goal is to offer therapeutic alternatives based on biology and genetic markers and to facilitate access to new therapies developed for adult AML in pediatric cases.

The manuscript is descriptive and does not present original data. It is a literature review followed by a perspective on future developments. The tables and figures provide a good overview. The perspectives focus heavily on therapies targeting genetic alterations or antigens. The authors discuss little about strategies considering measurable residual disease kinetics and possibly managing molecular relapses or persistent MRD. I have no further comments on the content.

Author Response

Thank you for your comment.

My best regards, 

Valeria Ceolin

Reviewer 2 Report

Comments and Suggestions for Authors

A very interesting and well-prepared manuscript, describes the history of AML  therapy, current methods used in Europe and around the world, and new directions in therapy, especially targeted treatment. I have no comments on the presented manuscript.

Author Response

Thank you.

My best regards, 

Valeria Ceolin